# Case of Congenital Hemolytic Anemia with *ATP11C* and *ANK1* Variants

**DOI:** 10.3390/children10101600

**Published:** 2023-09-25

**Authors:** Wei Xu, Mengmeng Ma, Sai Zhao, Yufang Yuan, Zhaofang Tian

**Affiliations:** 1The Department of Pediatrics, The Affiliated Huaian No.1 People’s Hospital of Nanjing Medical University, No 1 Huanghe Road, Huaiyin District, Huaian 223300, China; xuw0313@163.com (W.X.); hayytzf@njmu.edu.cn (Z.T.); 2The Department of Neonatology, The Affiliated Huaian No.1 People’s Hospital of Nanjing Medical University, No 1 Huanghe Road, Huaiyin District, Huaian 223300, China; mammeng@126.com (M.M.); zhaosai2008@126.com (S.Z.)

**Keywords:** *ATP11C*, *ANK1*, anemia, clinical genetics, DNA sequencing

## Abstract

A male infant of Han descent, with a G_1_P_1_ mother and gestational age of 40^+4^ weeks, was born via cesarean section owing to his mother having pregnancy complications, including premature rupture of membranes, chorioamnionitis, and gestational diabetes. On the first day after birth, routine blood examination showed that his total red blood cells count was 2.32 × 10^12^/L, hemoglobin count was 77 g/L, and C-reactive protein count was 48.99 mg/L. After receiving an anti-infection treatment for 10 days and two blood transfusions (100 mL in total), he was discharged from a neonatal intensive care unit (NICU). Accessory examinations showed that reticulocytes in the peripheral blood were significantly increased, the morphology of red blood cells was normal, and all hemolysis-related examinations were normal; bone marrow examinations showed that the proliferation of the red blood cell system was low and serum ferritin and vitamin B_12_ levels were elevated. Because of the unexplained hemolysis, a whole-exome sequencing examination was performed. The results showed a hemizygous variant of the *ATP11C* gene (c.3136a>t/p ile 1046phe) and a frame-shift variant of the *ANK1* gene (c.937del/pala313 leufs*19). After a six-month follow-up, the serum ferritin and vitamin B_12_ levels had gradually decreased to normal levels, and hemoglobin and reticulocyte values were 97 g/L and 7.17%, respectively, in the peripheral blood. No splenomegaly was found in physical examination.

## 1. Introduction

Anemia is the most common disease in newborns and children, and the causes of early neonatal anemia are complex. Congenital hemolytic anemia (CHA) is a rare hereditary disease characterized by an increased destruction of red blood cells (RBC). In CHA, the disruption of endogenous RBC enzymes, RBC membranes, and hemoglobin (HGB) can lead to hemolysis. The typical clinical manifestations are pallor, jaundice, and frequent splenomegaly. Laboratory features include anemia, hyperbilirubinemia, and reticulocytosis. In some cases of CHA, splenectomy is effective. At the same time, supportive care for blood transfusions is also a pillar of treatment [1].

The *ATP11C* gene is located on the human X chromosome and encodes a protein belonging to the p-type ATPase family, mainly responsible for regulating various biological processes such as growth and development, metabolism, and immunity [2,3].

The *ANK1* gene is located on chromosome 8 and encodes a RBC anchor protein [4]. The variation of the *ANK1* gene leads to changes in the structure of the encoding anchor protein, reducing its plasticity and stability on the RBC membrane. This accelerates the dissolution and destruction of RBC [5].

Here, we report a case with variants in both the *ATP11C* and *ANK1* genes. The proband is a term infant of Chinese Han ethnicity, characterized by a progressive aggravation of anemia, a suppressed proliferation of bone marrow RBC, peripheral blood reticulocytosis, and significant increase in serum iron and vitamin B_12_. We emphasize the diagnostic value of early-intervention genetic testing for neonatal anemia of unknown cause.

## 2. Case Presentation

### 2.1. Initial Presentation

The proband was a male infant of Han descent, with a G1P1 mother, who was born via cesarean section at 40^+4^ weeks owing to pregnancy complications, including chorioamnionitis, premature rupture of membranes, and gestational diabetes. The Apgar score was 9–10 and the birth weight was 3500 g. He was admitted within 1 day and 14 h of birth to the Department of Neonatology, the Affiliated Huaian No. 1 People’s Hospital of Nanjing Medical University, due to anemia and jaundice. Peripheral blood RBC count was 2.32 × 10^12^/L, HGB was 77 g/L, and CRP was 48.99 mg/L. The decrease in peripheral blood HGB in the child indicated severe anemia. CRP increased, suggesting the presence of infection.

The blood types of the proband’s father and mother were AB and O, respectively, and the proband’s mother had a history of anemia due to iron deficiency. No splenomegaly was found in the physical examination at admission. The results of the improved direct anti-human globulin test, antibody release test, and free antibody test were all negative. A hemagglutination report showed a PT of 19.9 s, prothrombin activity of 36.4%, and APTT of 47.7 s. The levels of GPT, GOT, and γ-GT and the blood electrolyte levels were within a normal range. Furthermore, the concentration of total bilirubin was 221.1 μmol/L, indirect bilirubin was 209.2 μmol/L, direct bilirubin was 11.9 μmol/L, LDH was 392.0 μ/L, CK was 721.0 μ/L, and CK-MB was 72.6 μ/L. The presence of hepatitis B markers, anti-HIV antibody, treponema pallidum antibody, and anti-hepatitis C antibody was also examined. In order to rule out the possibility of bleeding, chest X-ray photography, skull CT, and abdominal ultrasound were performed, but none showed any abnormality. A CSF examination (performed to rule out the possibility of central nervous system infection due to significantly increased levels of infection indicators) showed no abnormalities either. The results for anti-alkali HGB, HGB A2, trace HGB electrophoresis, isopropanol, erythrocyte permeability fragility test, H inclusion, denatured globossome, and glucose-6-phosphatase activity were all normal. The result of the methemoglobin reduction test was 87% (reference value > 75%). The plasma-free HGB concentration was 37.5 mg/L (reference value < 40 mg/L). Peripheral blood tests primarily showed a decrease in HGB (82 g/L) and an increase in reticulocytes (8.44%). Following six days of anti-infection treatment using cefepime, the levels of PCT and CRP in blood decreased from 49.916 ng/mL and 48.99 mg/L, respectively, to within normal ranges. The patient was discharged after 10 days of hospitalization, during which two blood transfusions were conducted.

### 2.2. Disease Progression and Further Treatment

At 33 days old, the proband was readmitted to the Affiliated Huaian No. 1 People’s Hospital of Nanjing Medical University after a routine blood examination showed a decrease in HGB concentration. The laboratory examination showed that RBC count was 1.98 × 10^12^/L, HGB count was 60 g/L, hematocrit was 18.2%, MCV was 102.3 fl, MCH was 30.3 pg, MCHC was 330 g/L, Ret was 4.67%, vitamin B_12_ concentration was 680 pmol/L (normal range 145–569 pmol/L), folic acid concentration was higher than 45.4 nmol/L, and ferritin concentration was 631 ng/mL (normal range, 30–400 ng/mL). Repeated low RBC count and HGB levels indicate the presence of severe anemia. Further, upon bone marrow puncture examination, images showed that granulocytes and megakaryocytes were proliferating actively, erythroid cells were proliferating less, and platelet clusters could also be seen (shown in Figure 1). Peripheral blood smear collected at the same time showed no significant increase or decrease in white blood cell count; however, the proportion of neutrophils decreased. The morphology of mature RBC was generally normal, and no nucleated RBC were observed in 100 white blood cells (shown in Figure 2). Following an 80 mL blood transfusion, the infant was discharged and followed up in an outpatient department. Because of the discordant or non-contributive clinical features and/or biological tests in this case, whole-exome sequencing (WES) was used by Guangzhou Jinyu Medical Inspection Group Co., Ltd. for further testing.

### 2.3. Whole-Exome Sequencing Results

WES revealed *ANK1* and *ATP11C* variants in the proband (as seen in the reference sequence of GRCh37). Sanger sequencing showed that the *ATP11C* gene was c.3136A>T (p.Ile1046Phe), a hemizygous variant (shown in Figure 3). Although the *ANK1* gene presented as c.937del (a frameshift variant of p.Ala313Leufs*19), as shown in Figure 4, a hemizygous variant could theoretically lead to a loss of normal protein function through nonsense-mediated mRNA degradation or premature termination of the coding amino acid sequence.

### 2.4. Follow-Up

At six months old, the patient’s total RBC count was 3.34 × 10^12^/L, the HGB count was 97 g/L, hematocrit was 27.8%, MCV was 83.2 fl, MCH was 29.0 pg, MCHC was 349 g/L, Ret was 7.17%, vitamin B_12_ level was 515 pmol/L, folate level was higher than 45.4 nmol/L, and ferritin level was 449.00 ng/mL.

As mentioned earlier, blood transfusion support is the mainstay of treatment for patients with congenital hemolytic anemia. The proband received an intervention of blood transfusion support treatment during the course of the illness, and the situation improved by the follow-up, which confirmed the effectiveness of blood transfusion treatment and suggested that his condition was under control. The increase in ferritin is considered to be related to blood transfusion, and, if necessary, iron removal treatment should also be given. In addition, the proband still needs regular follow-ups to prevent further changes in his condition.

## 3. Discussion

Congenital hemolytic anemia (CHA) is a group of rare genetic disorders characterized by increased destruction of RBCs. The clinical features of CHA vary from severe neonatal or even prenatal anemia with high morbidity and transfusion dependence to well-compensated hemolysis without anemia [6]. Next-generation sequencing (NGS) allows for massive parallel sequencing of numerous genes and therefore provides a very suitable approach for the genetic dissection of CHA. Lamisse used WES to explore 40 CHA patients, of whom one patient carried both a likely pathogenic hemizygous variation in *ATP11C* (c.2434C>T) and a variant of uncertain significance in *ANK1* (c.4558G>C). This patient’s blood smear was normal and ektacytometry showed an atypical profile with only dehydration and no change in osmotic resistance [7]. The proband in this current article was a term infant of Chinese Han ethnicity, characterized by a progressive aggravation of anemia, suppressed proliferation of bone marrow RBC, peripheral blood reticulocytosis, and significant increase in serum iron and vitamin B_12_ levels. Through WES, an *ATP11C* gene hemizygous variant (c.3136A>T/p.Ile1046Phe)) and an *ANK1* gene frameshift variant (c.937del/pAla313Leufs*19) were identified.

The *ATP11C* gene is located at 138,808,505,138,914,447 bp of the human X chromosome and consists of 105,943 bases. The gene encodes a protein that belongs to the P-type ATP enzyme family and comprises 1129 amino acids. This protein is uniquely expressed and transports multiple ions to regulate metabolic activities within the cell. Additionally, it mediates the transport of large amounts of phospholipids. As a lipid flippase, it transports phospholipids from cell to cell, i.e., it enriches extracellular phospholipids within the cell, which may play a crucial role in modulating various biological processes such as growth and development, metabolism, and immunity [2,3]. ATP11C is a member of the P4-ATP enzyme family and a major phosphatidylserine (PS) flippase located on the plasma membrane. ATP11C deficiency causes a defect in B-cell maturation, anemia, and hyperbilirubinemia. ATP11C-b regulates PS distribution in distinct regions of the plasma membrane in polarized cells [8]. *ATP11C* has also been reported as a candidate gene for hypoparathyroidism [9]. Variants in *ATP11C* cause defects in lymphatic B cell differentiation [10] and induce bile stasis [11].

*ATP11C* maintains the asymmetry of phospholipids in erythrocytes during phospholipid transport. Impaired phospholipid transport across the cell membrane can result in anemia. The expression of PS on the cell membrane of aging RBC is regarded as a signal of phagocytosis by macrophages. Many factors, including alterations in intracellular substances and a decrease in ATP11C levels, lead to decreased phospholipase activity in aging RBC, which in turn leads to the expression of PS on cell membranes [12]. In mice, the *ATP11C* variants has been found to result in morphological alterations in and a reduced life span of RBC. Although the RBC of *ATP11C*-deficient mice develop normally, the mature RBC exhibit an abnormal shape (increased oral cells) and a lifespan that is shorter than that in control mice from the same litter, leading to anemia [13]. In patients with sickle cell anemia, the ratio of ATP11C/phospholipid scramblase 1 (PLSCR1) or ATP11C and PLSCR1 (if analyzed separately) influences the risk of acute or chronic organ injury, the equilibrium between ATP11C and PLSCR1 in SCA, and how or whether they lead to the development of acute or chronic clinical manifestations [14].

In 2016, Nobuto Arashiki et al. first reported that ATP11C is the primary flippase in human RBC, based on an analysis of a patient with mild hemolytic anemia. They also showed that variants in the *ATP11C* gene result in congenital mild hemolytic anemia with X-linked recessive inheritance [15]. A more recent case study by Zhang, H. M. et al. also reported similar findings [16]. Unlike the present case, patients in the aforementioned two cases were diagnosed at the ages of 13 and 15, respectively, which may be associated with relatively mild anemia symptoms. In the present case, the infant developed severe anemia immediately after birth, necessitating several blood transfusions in the short term. However, a bone marrow examination of the proband conducted one month after birth showed low erythroid hyperplasia, which was inconsistent with the active bone marrow hyperplasia commonly observed in hemolytic anemia. Furthermore, this observation was inconsistent with the significant increase in reticulocytes in the peripheral blood of the infant. The apparent increase in serum ferritin levels in pediatric patients may be associated with repeated blood transfusions. However, the reason for the increase in vitamin B_12_ levels remains unclear, and further investigation is required to determine whether it is associated with double gene variants. Although hemolytic anemia due to variants in the *ATP11C* gene is associated with spleen macrophages, splenomegaly has not been reported in documented cases, and it is yet to be determined whether this clinical feature is associated with the disease. It also remains to be elucidated whether it is associated with double gene variants. Although the variants in *ATP11C* that cause hemolytic anemia are associated with macrophages in the spleen, none of the previously documented cases have reported an enlargement of the spleen. Whether this clinical feature is associated with the disease is yet to be determined. Extensive phenotypic support will be necessary to classify the *ATP11C* variants in this child as pathogenic.

The *ANK1* gene is located on chromosome 8 (P11.2) and encompasses 42 exons. The gene encodes an erythrocyte anchor protein that is primarily expressed on the erythrocyte membrane through its association with anion channels, the Rh complex, and contracted ovalbumin. Variants in the *ANK1* gene result in alterations in the structure of the encoded anchor protein, thereby reducing its plasticity and stability on the erythrocyte membrane. This phenomenon accelerates the dissolution and destruction of erythrocytes. MCHC levels higher than 365 g/L or an MCHC/MCV ratio greater than 0.36 in the blood are valuable diagnostic indices for neonatal Hereditary Spherocytosis(HS). The median age at which splenomegaly occurs is six years old; however, a few studies have highlighted that some cases of newborns with HS may exhibit a nonspecific phenotype. Additionally, certain diagnostic indicators in the guidelines may not apply to the diagnosis of newborns with HS [4]. The *ANK1* variant appears to be pathogenic. The osmotic fragility test is usually negative until 6 months of age and it is not uncommon to see only a few spherocytes in peripheral blood. Newborns also have immature spleen function, but this is unlikely to cause hemolysis due to clearance of PS-expressing RBC. Establishing a phenotypic diagnosis of HS in newborns is difficult and multiple tests, including some delayed testing, are necessary at a later period. This indicates that, in the present case, the role of the variants in the *ANK1* gene cannot be ruled out.

## 4. Conclusions

In conclusion, we report for the first time a neonatal case of congenital hemolytic anemia with variants in both the *ATP11C* and *ANK1* genes, characterized by severe anemia after birth, low erythroid hyperplasia in the bone marrow, and an apparent increase in the levels of ferritin and vitamin B_12_ in serum. These findings highlight the importance of further investigations into similar cases in clinical settings.

## Figures and Tables

**Figure 1 children-10-01600-f001:**
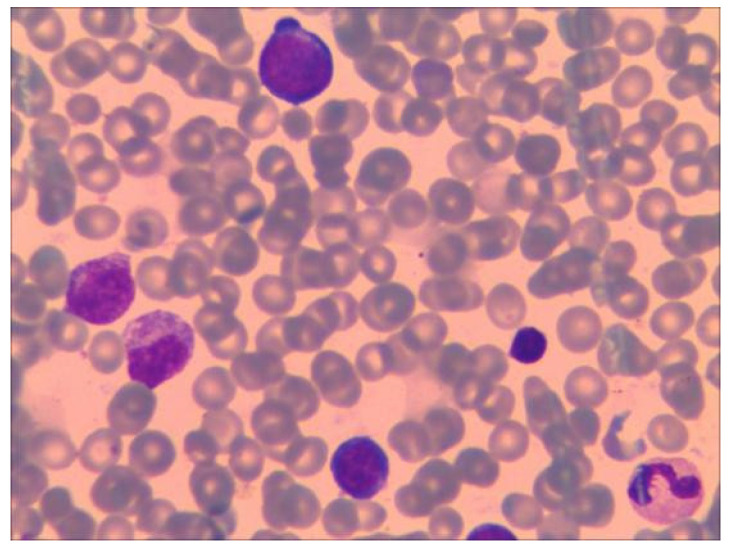
Bone marrow examination. Granulocytes and megakaryocytes proliferate actively, erythroid cells proliferate less, and platelet clusters can be seen.

**Figure 2 children-10-01600-f002:**
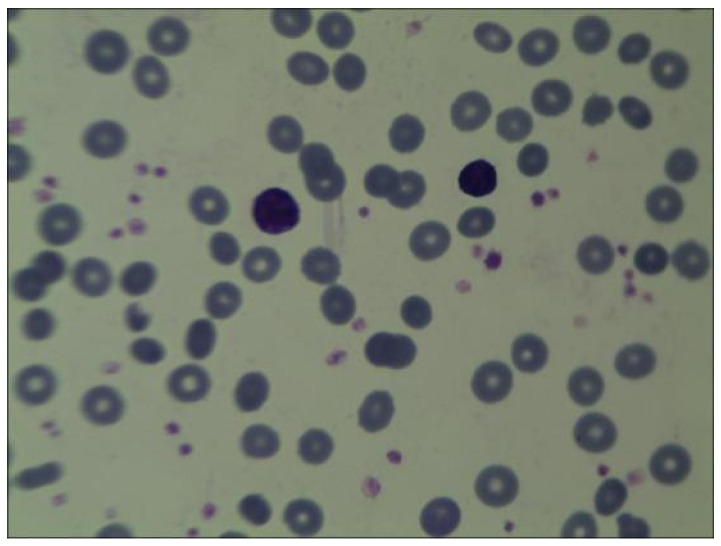
Peripheral blood examination. The morphology of mature RBC is normal, with no significant increase or decrease in white blood cells. The morphology of granulocytes, lymphocytes, and monocytes is normal, and platelets are scattered and visible in clusters.

**Figure 3 children-10-01600-f003:**
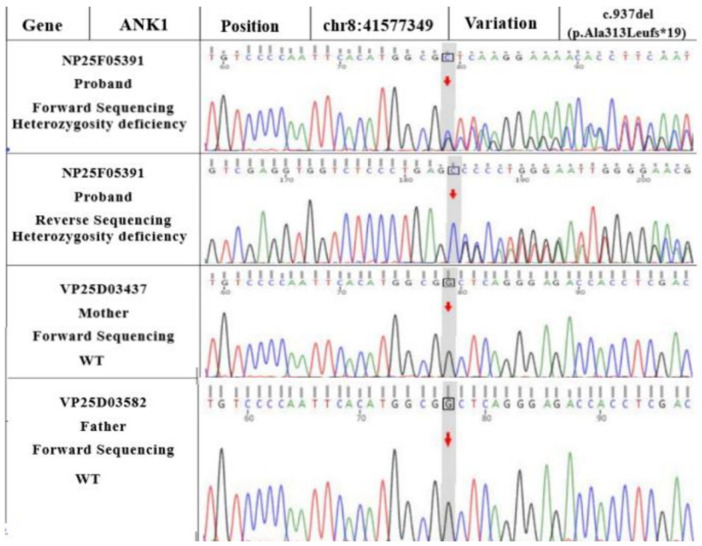
*ANK1* gene variants in WES. The c.937del (p.Ala313Leufs*19) carried by the proband is a frameshift variant caused by non-triple base deletion in the coding region of the *ANK1* gene. This variant was confirmed via Sanger sequencing. This variant was not reported in the literature or in the large-scale population frequency database gnomAD. (A: adenosine; T: thymine; C: cytosine; G: guanine).

**Figure 4 children-10-01600-f004:**
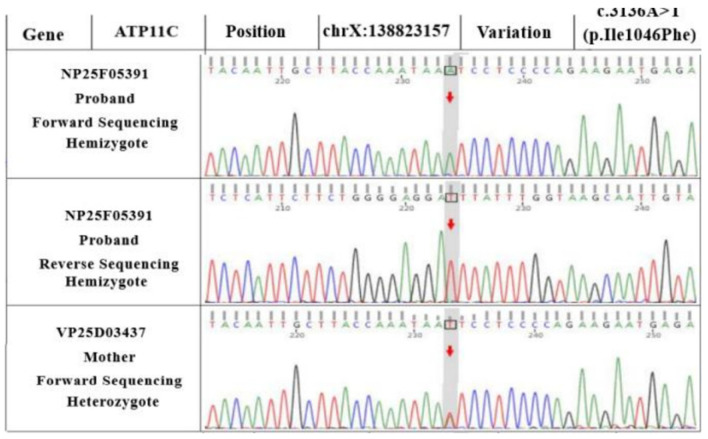
*ATP11C* gene variants in WES. The c.3136A>T (p.Ile1046Phe) variant carried by the proband was a missense variant in the coding region of the *ATP11C* gene. The variant was inherited from the mother, as verified via Sanger sequencing. This variant was not reported in the literature or in the large-scale population frequency database gnomAD.(A: adenosine; T: thymine; C: cytosine; G: guanine).

## Data Availability

All data generated or analyzed during this study are included in this article. Further inquiries can be directed to the corresponding author.

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
