# Peer review of "Case of Congenital Hemolytic Anemia with ATP11C and ANK1 Variants"

_children, 2023, doi:10.3390/children10101600_

Round 1
Reviewer 1 Report
The present manuscript entitled "Case of congenital hemolytic anemia with ATP11C and ANK1 mutations" presented the case of a newborn of Chinese Han ethnicity. He had some clinical manifestations such as anemia, suppression of proliferation of bone marrow red blood cell system, peripheral blood reticulocytosis, etc. The investigations performed to understand the cause of these manifestations allowed the authors to identify two mutations in ATP11C and ANK1 genes. Then, they suggested the relationship of these mutations with severe anemia after birth, low erythroid hyperplasia in the bone marrow, and an apparent increase in serum ferritin and vitamin B12 levels.
However, the report presents some inconsistencies in the data, principally those related to the sequencing.
1- I do not understand the electropherogram of Figure 3, which is supposed to show the frameshift mutation, c.937del(p.Ala313Leufs*19), caused by a non-triple base deletion in the coding region of the ANK1 gene.
We have:
Father and Mother: TGT CCC CAA TTC ACA TGG CG"G" CTC AGG GAG ACC ACC TCG AC
Proband - F: TGT CCC CAA TTC ACA TGG CG"C" ...................
Proband - R: GTC GAG GTG GTC TCC CTG AG"C" ....................
Although the deletion, at the position depicted by the ", in one of the child's ANK1 alleles leads to a frameshift, the reverse nucleotide sequence before this position does not match the others. Furthermore, the reverse sequencing of the proband seems different from the forward sequencing.
2- We have the same problem with the ATP11C gene (Figure 4). Except for the mutation, the reverse sequence of the proband is different from the forward sequences of the proband and mother (TCT CAT TCT TCT GGG GAG GA"T" TTA TTT GGT AAG AAT TGT A versus TAC AAT TGC TTA CCA AAT AA"A/T" TCC TCC CCA GAA GAA TGA GA, respectively).
3- I think there is a mistake about the localization of the ATP11C gene (confer the first sentence of the discussion). Furthermore, the second sentence, which reports its localization in the mouse, is unnecessary.
4- On page 3, the authors state that the mutation identified in the ATP11C gene has not been previously reported and that, based on the available evidence, the clinical significance of this variant is unclear. I would like to know if this evidence is related to this variant. If so, it is strange, as they mention that it is the first time that the variant was reported.
5- In the first paragraph of the discussion section, the authors stated the following: ATP11C leads to B cell maturation defects, anemia, and hyperbilirubinemia. Do you want to mean that defects in ATP11C lead to defects in B cell maturation, anemia, etc.? If so, this sentence needs to be clarified.
These are just some of the issues that need to be solved.
The manuscript presents some English spelling issues that need to be checked and solved.
Author Response
Dear Editor and reviewer:
Thank you very much for giving us an opportunity to revise our paper. Based on your valuable suggestions and reviewers’ comments, we have revised our original manuscript (article ID: children-2610346). We will respond to your questions point to point.
I hope you give us a positive response for the publication of this paper.
Thank you for taking time for consideration.
Dr.Zhaofang Tian
Question1- I do not understand the electropherogram of Figure 3, which is supposed to show the frameshift mutation, c.937del(p.Ala313Leufs*19), caused by a non-triple base deletion in the coding region of the ANK1 gene.
We have:
Father and Mother: TGT CCC CAA TTC ACA TGG CG"G" CTC AGG GAG ACC ACC TCG AC
Proband - F: TGT CCC CAA TTC ACA TGG CG"C" ...................
Proband - R: GTC GAG GTG GTC TCC CTG AG"C" ....................
Although the deletion, at the position depicted by the ", in one of the child's ANK1 alleles leads to a frameshift, the reverse nucleotide sequence before this position does not match the others. Furthermore, the reverse sequencing of the proband seems different from the forward sequencing.
Our answer: Thanks very much for your carful checking.
The results of forward sequencing and reverse sequencing show a reverse complementary relationship.Therefore,in the mutation results of ATP11C, to check whether the reverse sequencing results match the forward sequencing results, the sequence should be reversed and converted into complementary bases for comparison. For example,the proband's reverse sequencing result “CTCATTCTTCTGGGGAGGATTTATTTGGTAAGCAATTGTA”should be converted into “TACAATTCGTTACCAAATAA"A"TCCTCCCCAGAAGAATGAGA”,then you will find that it is consistent with the results of forward sequencing.
Question2- We have the same problem with the ATP11C gene (Figure 4). Except for the mutation, the reverse sequence of the proband is different from the forward sequences of the proband and mother (TCT CAT TCT TCT GGG GAG GA"T" TTA TTT GGT AAG AAT TGT A versus TAC AAT TGC TTA CCA AAT AA"A/T" TCC TCC CCA GAA GAA TGA GA, respectively).
Our answer: As that answer to the first question say.when base deletion mutation is concerned, the difference between forward and backward sequencing is even greater.These involve many technical problems.
Question3-I think there is a mistake about the localization of the ATP11C gene (confer the first sentence of the discussion). Furthermore, the second sentence, which reports its localization in the mouse, is unnecessary.
Our answer:In the revised version,We have revised a mistake about the localization of the ATP11C gene,and deleted the content related to the mouse.
Question4- On page 3, the authors state that the mutation identified in the ATP11C gene has not been previously reported and that, based on the available evidence, the clinical significance of this variant is unclear. I would like to know if this evidence is related to this variant. If so, it is strange, as they mention that it is the first time that the variant was reported.
Our answer:In the revised version,We have deleted related content.
Question5- In the first paragraph of the discussion section, the authors stated the following:ATP11C leads to B cell maturation defects,anemia,and hyperbilirubinemia.Do you want to mean that defects in ATP11C lead to defects in B cell maturation, anemia, etc.? If so, this sentence needs to be clarified.
Our answer:In the revised version,We have revised related content.Expressed as“ATP11C deficiency causes a defect in B-cell maturation, anemia and hyperbilirubinemia.”

Reviewer 2 Report
The case study highlights a combination of mutations in ATP11C and ANK1 genes correlated to congenital hemolytic anemia. The description and discussion of the case are well thought out and well written. I have the following suggestions or comments about the study:
1. Proband is referred to in present tense in the abstract and past tense in the introduction, unsure about this writing choice.
2. There are several formatting errors with bold text, different formats, larger fonts, etc.
3. ATP11C location does not match with the ensemble database (it is mentioned that it is on Chromosome X: 139,726,346-139,945,276 reverse strand. GRCh38:CM000685.2) Authors should confirm the location and refer to the version of the human genome they are citing (references are completely missing for this part of the discussion).
4. I don't understand the need to cite the nationality of scholars stating, 'Japanese scholars' or 'Chinese scholars' etc. Wouldn't it be better to cite the study directly (e.g., author et al., year, journal)
5. Authors claim to be reporting the first case of congenital hemolytic anemia with concurrent ATP11C and ANK1 mutations, however, a case (P40) from the following 2020 study has reported this combination of mutations:
Mansour-Hendili L, Aissat A, Badaoui B, Sakka M, Gameiro C, Ortonne V, Wagner-Ballon O, Pissard S, Picard V, Ghazal K, Bahuau M, Guitton C, Mansour Z, Duplan M, Petit A, Costedoat-Chalumeau N, Michel M, Bartolucci P, Moutereau S, Funalot B, Galactéros F. Exome sequencing for diagnosis of congenital hemolytic anemia. Orphanet J Rare Dis. 2020 Jul 8;15(1):180. doi: 10.1186/s13023-020-01425-5. PMID: 32641076; PMCID: PMC7341591.
It would be wiser if the authors were more specific about this case and defined it as a first-ever case in a different manner.
The quality of English is ok.
Author Response
Dear Editor and reviewer:
Thank you very much for giving us an opportunity to revise our paper. Based on your valuable suggestions and reviewers’ comments, we have revised our original manuscript (article ID: children-2610346). We will respond to your questions point to point.
I hope you give us a positive response for the publication of this paper.
Thank you for taking time for consideration.
Dr.Zhaofang Tian
Question1Proband is referred to in present tense in the abstract and past tense in the introduction, unsure about this writing choice.
Our answer: Thanks very much for your carful checking,We have deleted related content in the revised version.
Question2There are several formatting errors with bold text, different formats, larger fonts, etc.
Our answer: Thanks very much for your carful checking,We have revised relatedmistakes in the revised version.
Question3ATP11C location does not match with the ensemble database (it is mentioned that it is on Chromosome X: 139,726,346-139,945,276 reverse strand. GRCh38:CM000685.2) Authors should confirm the location and refer to the version of the human genome they are citing (references are completely missing for this part of the discussion).
Our answer: In this paper, the reference sequence of GRCh37 is used instead of GRCh38.
Question4I don't understand the need to cite the nationality of scholars stating, 'Japanese scholars' or 'Chinese scholars' etc. Wouldn't it be better to cite the study directly (e.g., author et al., year, journal)
Our answer:We use the author's name instead of nationality when quoting references in the revised edition.
Question5Authors claim to be reporting the first case of congenital hemolytic anemia with concurrent ATP11C and ANK1 mutations, however, a case (P40) from the following 2020 study has reported this combination of mutations:
Our answer:Thank you for reminding us.There is an insufficient in our literature retrieval.We will add the content of the reference to the first paragraph of discussion in the revised edition, and modify“we report for the first time a case of congenital hemolytic anemia with mutations in both the ATP11C and ANK1 genes” as “we report for the first time a neonatal case of congenital hemolytic anemia with mutations in both the ATP11C and ANK1 genes”in the last paragraph.

Reviewer 3 Report
Dear Authors,
I have read a case report describing a case of congenital hemolytic anemia (from on, I abbreviate it as CHA) with two gene mutations, namely ATP11C and ANK1. I have several reservations
1) The abstract is too short and does not capture any significant importance
2) The manuscript is overall very descriptive, without showing the logic of clinical thinking behind it. How did the authors arrive at CHA? At which point of the examination did it show? Why were so many examinations checked? What were the clinical presentations? What were the original levels of Hb and other basic RBC indices? What other therapies were given during hospitalization besides antibiotics and blood transfusions?
3) What was the clinical follow-up after the genetic testing?
4) The discussion heavily discussed the genetical profiles without any other mention on the overall CHA diagnosis...
The English is a mess, to put it lightly. Grammatical errors and spelling mistakes make the manuscript very hard to read. Some mistakes feel like they are being translated directly from Chinese "Gnranlar" or "Granuid", which make no sense
Author Response
Dear Editor and reviewer:
Thank you very much for giving us an opportunity to revise our paper. Based on your valuable suggestions and reviewers’ comments, we have revised our original manuscript (article ID: children-2610346). We will respond to your questions point to point.
I hope you give us a positive response for the publication of this paper.
Thank you for taking time for consideration.
Dr.Zhaofang Tian
Question 1) The abstract is too short and does not capture any significant importance?
Our answer:We have perfected the content of the abstract and added the necessary information in the revised version.
Question2 The manuscript is overall very descriptive, without showing the logic of clinical thinking behind it. How did the authors arrive at CHA? At which point of the examination did it show? Why were so many examinations checked? What were the clinical presentations? What were the original levels of Hb and other basic RBC indices? What other therapies were given during hospitalization besides antibiotics and blood transfusions?
Our answer:In the revised edition, we have annotated and revised the relevant Laboratory and instrument inspection, added the contents of blood routine at the first admission and supplemented the content of follow-up.In this case, the patients were mainly treated with anti-infection,blood transfusion and symptomatic treatment.
Question3 What was the clinical follow-up after the genetic testing?
Our answer: The clinical follow-up after the genetic testing was in “1.4 Follow up.”
Question4The discussion heavily discussed the genetical profiles without any other mention on the overall CHA diagnosis...
Our answer:We added relevant content in the first paragraph of the discussion.

Round 2
Reviewer 1 Report
I thank the authors for their efforts to improve the quality of the manuscript and for the reply. Unfortunately, the manuscript still needs to be reorganized. Ideas in some parts of the manuscript seem confusing. In my opinion, being a case report, it is unnecessary to divide the manuscript into sections (introduction, methods, results, discussion, and conclusion) as the result section is your introduction, which presents the case, reporting the laboratory and clinical data of the child and its parents. I also do not know if the abstract is necessary. Please carefully evaluate the right way to publish this data. It may be in a Brief report or Case report form.
What will be the concrete action that can be done based on the findings? I ask that because I felt a gap between 1) the clinical and laboratory data that led to the investigation of the mutation, 2) the mutations identified, 3) the clinical care (treatments, etc.) performed based on these findings, and 4) the data clinical evolution of the child after six months. Did you do a specific care based on the mutations identified?
Furthermore, the manuscript needs to be submitted to an English native speaker.
Author Response
Dear Editor and reviewer:
Thank you very much for giving us an opportunity to revise our paper. Based on your valuable suggestions and reviewers’ comments, we have revised our original manuscript. All amendments are highlighted in the manuscript and a list of answers is included in this letter.
I hope you give us a positive response for the publication of this paper.
Thank you for taking time for consideration.
Dr. Zhaofang Tian
Major:
Question 1: In my opinion, being a case report, it is unnecessary to divide the manuscript into sections (introduction, methods, results, discussion, and conclusion) as the result section is your introduction, which presents the case, reporting the laboratory and clinical data of the child and its parents. I also do not know if the abstract is necessary. Please carefully evaluate the right way to publish this data. It may be in a Brief report or Case report form.
Our answer: Thank you very much for your suggestion.In the revised version,We have revise the article according to the format of “case report” of Children .
Question 2: What will be the concrete action that can be done based on the findings?
Our answer: Thank you for your suggestion.In the revised version,We have added some contenent in the section of Follow-up .

Reviewer 3 Report
Dear Authors,
While there is some effort towards improvement, there is still a lot of points that need to be addressed:
1) There is a lack of introduction, and the section seems off. 1.1, 1.2 and so on lacks meaning as there is no header.
2) The positioning of the sentence seems awkward and there is a lack of clarity
3) The authors still cite numbers and hoping the readers will get it. The authors should explain what the values mean, and what is the logical next step in the clinical picture. While there is some attempt in improving this logic, it is still largely deficient.
The English language has not been improved at all, and after the revision, it has become worse.
Author Response
Dear Editor and reviewer:
Thank you very much for giving us an opportunity to revise our paper. Based on your valuable suggestions and reviewers’ comments, we have revised our original manuscript. All amendments are highlighted in the manuscript and a list of answers is included in this letter.
I hope you give us a positive response for the publication of this paper.
Thank you for taking time for consideration.
Dr. Zhaofang Tian
Major:
Question 1: There is a lack of introduction, and the section seems off. 1.1, 1.2 and so on lacks meaning as there is no header.
Our answer: Thank you very much for your suggestion.We have improved the relevant content in the article.
Question 2: The positioning of the sentence seems awkward and there is a lack of clarity.
Our answer: Thank you for your suggestion. We have selected the editing services from MDPI's Author Services for the article.
Question 3: The authors still cite numbers and hoping the readers will get it. The authors should explain what the values mean, and what is the logical next step in the clinical picture. While there is some attempt in improving this logic, it is still largely deficient.
Our answer: Thank you very much for your suggestion.We have explained the relevant values and improved the relevant content in modified version.
Comments on the Quality of English Language:The English language has not been improved at all, and after the revision, it has become worse.
We have selected the editing services from MDPI's Author Services for the article.

Round 3
Reviewer 3 Report
Dear Authors,
Thank you for persevering through the fourth round of revision. I only have one last comment, please include the reference and citation for the following paragraph:
"The ATP llc gene is located on the human X chromosome and encodes a protein belonging to the p-type ATPase family, mainly responsible for regulating various biological processes such as growth and development, metabolism, and immunity.
The ANK1 gene is located on chromosome 8 and encodes a red blood cell anchor protein. The mutation of the ANK1 gene leads to changes in the structure of the encoding anchor protein, reducing its plasticity and stability on the red blood cell membrane. This accelerates the dissolution and destruction of red blood cells."
The rest of the manuscript has been improved.
The English language has been largely improved.
Author Response
Dear Editor and reviewer:
Thank you very much for giving us an opportunity to revise our paper. Based on your valuable suggestions and reviewers’ comments, we have revised our original manuscript. All amendments are highlighted in the manuscript and a list of answers is included in this letter.
I hope you give us a positive response for the publication of this paper.
Thank you for taking time for consideration.
Dr. Zhaofang Tian
Major:
Question 1: I only have one last comment, please include the reference and citation for the following paragraph
Our answer: Thank you very much for your suggestion,We have added references and citation for the related paragraph.
